# Azido-Ceramides, a Tool to Analyse SARS-CoV-2 Replication and Inhibition—SARS-CoV-2 Is Inhibited by Ceramides

**DOI:** 10.3390/ijms24087281

**Published:** 2023-04-14

**Authors:** Daniela Brenner, Nina Geiger, Jan Schlegel, Viktoria Diesendorf, Louise Kersting, Julian Fink, Linda Stelz, Sibylle Schneider-Schaulies, Markus Sauer, Jochen Bodem, Jürgen Seibel

**Affiliations:** 1Institute of Organic Chemistry, Julius-Maximilians-Universität Würzburg, 97074 Würzburg, Germany; daniela.brenner@uni-wuerzburg.de (D.B.); louise.kersting@uni-wuerzburg.de (L.K.);; 2Institute of Virology and Immunobiology, Julius-Maximilians-Universität Würzburg, 97078 Würzburg, Germany; nina.geiger@stud-mail.uni-wuerzburg.de (N.G.); viktoria.diesendorf@stud-mail.uni-wuerzburg.de (V.D.);; 3Department of Biotechnology and Biophysics, Julius-Maximilians-Universität Würzburg, 97074 Würzburg, Germany; jan.schlegel@scilifelab.se (J.S.); m.sauer@uni-wuerzburg.de (M.S.)

**Keywords:** ceramides, SARS-CoV-2, azido-ceramides, sphingolipids

## Abstract

Recently, we have shown that C6-ceramides efficiently suppress viral replication by trapping the virus in lysosomes. Here, we use antiviral assays to evaluate a synthetic ceramide derivative α-NH2-ω-N3-C6-ceramide (AKS461) and to confirm the biological activity of C6-ceramides inhibiting SARS-CoV-2. Click-labeling with a fluorophore demonstrated that AKS461 accumulates in lysosomes. Previously, it has been shown that suppression of SARS-CoV-2 replication can be cell-type specific. Thus, AKS461 inhibited SARS-CoV-2 replication in Huh-7, Vero, and Calu-3 cells up to 2.5 orders of magnitude. The results were confirmed by CoronaFISH, indicating that AKS461 acts comparable to the unmodified C6-ceramide. Thus, AKS461 serves as a tool to study ceramide-associated cellular and viral pathways, such as SARS-CoV-2 infections, and it helped to identify lysosomes as the central organelle of C6-ceramides to inhibit viral replication.

## 1. Introduction

While many SARS-CoV-2 studies focus on targeting viral proteins such as the polymerase [1], the spike protein [2,3], or the main protease [4,5,6,7], fewer address the possibility of intervening in the host metabolism to impede virus entry or replication [8,9,10]. In this context, sphingolipid metabolism is a promising target because it has been revealed to be crucial for many viral and bacterial infections [11,12,13,14,15].

Sphingolipids are major components of cellular membranes in various organisms such as mammals, plants, and bacteria [16]. These lipids regulate membrane fluidity, polarity, and deformation, but also vesicle formation and sorting [14,17]. They also accumulate in membranes of the multivesicular body compartment, where their metabolic pathway regulates the formation of intraluminal vesicles (ILVs), thereby assisting in sorting, degrading, and recycling proteins [14,18,19]. Because they also regulate signal transduction, sphingolipids modulate cellular processes, including proliferation, differentiation, apoptosis, and membrane trafficking [14,20,21,22].

The sphingolipid metabolism is a complex network of ana- and catabolic pathways. The up- or downregulation of enzymes and sphingolipid concentrations significantly impacts overall membrane biology and, consequently, the pathogenesis of infections [11,23,24]. This includes biophysical membrane properties but also the regulation of pathogen uptake by sorting of receptors or enhancement of endocytosis, or, for bioactive sphingolipids, cellular signaling. For instance, this applies to ceramide, a critical intermediate of the sphingolipid metabolism that can undergo several transformations: headgroup modifications to yield sphingomyelin or glycosyl ceramides or the cleavage of the fatty acid to produce sphingosine.

Supporting the importance of sphingolipid metabolism for SARS-CoV-2 infection, viral entry was found to be sensitive to pharmacologic or genetic targeting of acid sphingomyelinase (ASM), mainly located in lysosomes [25,26]. By lysosomal fusion with the plasma membrane, this enzyme is also exposed on the cell surface, generating ceramides from sphingomyelin. These ceramides can form lipid rafts, enabling viral entry [26,27,28].

## 2. Results and Discussion

Recently we have shown that inhibition of the ceramidase raised ceramide concentration and inhibited SARS-CoV-2 replication, similar to treatment with short-chain C6-ceramide [29]. However, the subcellular location of the C6-ceramide in SARS-CoV-2 permissive cells was not determined. Thus, it is still unclear whether the ceramide blocked SARS-CoV-2 replication in the same compartment. Therefore, we used an α-NH_2_-ω-N_3_-C6-ceramide derivative (AKS461), similar to the C6-ceramide regarding chain length (Figure 1A) [30]. This ceramide is equipped with an azido and an additional amino group and can be used as a visualization probe since the azido group enables click-labeling with a fluorophore for super-resolution microscopy, while the amino group is applicable for fixation and expansion microscopy [30]. The amino group contributes to the lower mobility observed in fixed cells of the α-NH_2_-ω-N_3_-C6-ceramide in the membrane compared to the ω-N_3_-C6-ceramide [30]. We previously described the synthesis of AKS461 [30]. However, as the purification of AKS461 is challenging due to the formation of side products with similar characteristics, our original protocol was further optimized and now includes chromatography on deactivated silica gel.

First, we analyzed whether AKS461 would suppress SARS-CoV-2 similarly to C6-ceramides. Since we and others have described that compounds inhibit SARS-CoV-2 replication in a cell line-specific way, three different cell lines were analyzed [7]. Thus, the cytotoxicity of AKS461 was analyzed by determining the cell growth rate after 72 h in Vero and Huh-7 cells as described before [7,29]. Similarly, MTT assays were performed after 72 h in Calu-3 cells since automated cell counting failed with this cell line [7]. AKS461 showed cytotoxicity at 30 µM in these cell lines. Thus, a concentration of 7.5 µM was chosen.

To analyze if the influence of AKS461 on viral replication is similar to C6 ceramides, the cells were treated with AKS461 and infected with SARS-CoV-2 (MOI = 1). Cell culture supernatants were harvested 72 h after infection (Figure 1B). AKS461 inhibited viral replication 2.5 orders of magnitude in Vero, 1.8 in Huh-7 at 30 µM, and 0.7 in Calu-3 cells at 7.5 µM. To confirm these results, Vero cells were treated with 7.5 µM AKS461, similar to the most sensitive Calu-3 cell line, and subsequently infected with SARS-CoV-2, and viral RNA in the cell was visualized using super-resolution structured illumination microscopy (SIM) as previously described [29,31]. In DMSO-treated cells, viral RNAs appeared in distinct clusters (Figure 1C). Application of AKS461 decreased the fluorescence signal of CoronaFISH-Cy5, indicating that this derivative has a similar impact on viral replication as C6-ceramide. Additionally, the clustering of viral RNAs was reduced and more evenly distributed in the cell. Our results suggest that AKS461 suppresses SARS-CoV-2, similar to C6-ceramides. This underlines the importance of controlling ceramide amounts for viral replication.

After metabolic incorporation of the lipid by cells, the azido group can react in vivo with a fluorescent dye bearing a strained alkyne in a copper-free click-reaction. Serving as a surrogate for its natural counterpart, it enables to follow subcellular trafficking of the derivative and its metabolites. We treated Vero cells with 7.5 µM AKS461 for 24 h, similar to the CoronaFISH-Cy5 experiments. Then the cells were fixed and clicked with DBCO-BODIPY. The lysosomes were co-stained with a lyso-tracker (Figure 2). The fluorescence images demonstrate strong colocalization of AKS461 with lyso-tracker, indicating enrichment of the compound in lysosomes. Furthermore, since SARS-CoV-2 maturation and egress are strongly linked to lysosomal membranes [29,32,33], the accumulation of AKS461 in this organelle might directly influence SARS-CoV-2 replication.

Our data demonstrate that AKS461 is a valuable tool for studying short-chain ceramide analogs in cell biology. Its unique chemical structure renders it an efficient live-cell probe for advanced microscopy techniques to elucidate mechanistic details of endolysosomal trafficking in the context of viral entry. Furthermore, we were able to confirm with AKS461 that ceramides inhibit SARS-CoV-2 replication.

## 3. Materials and Methods

### 3.1. Synthesis

#### General Experimental Information

Commercially available chemical reagents, purchased from Sigma Aldrich (Saint Louis, MO, USA), Alfa Aesar (Haverhill, Lancashire, UK), TCI (Chuo-ku, Tokyo, Japan), Avanati Polar Lipids (Alabaster, AL, USA), and ACROS (Geel, Belgium), were used as received without further purification. All solvents were distilled before usage. Dry solvents were either dried using standard procedures or taken from Solvents Purification Systems from Inert. Moisture-sensitive reactions were performed under a dry nitrogen atmosphere. Analytical thin layer chromatography (TLC) was carried out with silica gel pre-coated aluminum plates with a thickness of 0.2 mm. The compounds were visualized with a ninhydrin stain solution (600 mg ninhydrin, 6 mL glacial acetic acid, 200 mL n butanol). Liquid column chromatography purification was performed with silica gel 60 (40–63 µm mesh, Macherey-Nagel, Düren, Germany).

Preparation of deactivated silica gel: K_2_CO_3_ was ground to a fine powder. Silica gel and 5–10 wt% of K_2_CO_3_ powder were mixed and refluxed in MeOH for 4 h. The mixture was filtered, and the silica gel was washed with MeOH several times and dried. AKS461 was synthesized analogous to the literature with a modified purification protocol due to the difficult purification of the target compound [30]. The sphingosine was coupled with 6-azido-*N*-Boc-*l*-norleucine, forming an *N*-Boc-protected derivative of a C6-ceramide. The purification of the crude product was performed on deactivated silica with a solvent system of CHCl_3_/MeOH (9:1). Then, the *N*-Boc group was cleaved with trifluoroacetic acid to form AKS461. In our former purification procedure, we had to add aqueous ammonia to the solvent system for column chromatography. By using deactivated silica and a solvent system CH_2_Cl_2_/MeOH (20:1), we could isolate AKS461 more easily and remove by-products and remaining cytotoxic educts.

Nuclear magnetic resonance (NMR) spectra were recorded on a Bruker Avance (Billerica, MA, USA) III HD 400/600 at 295 K. Chemical shifts (δ) are given in parts per million (ppm) in reference to the solvent residual proton signal (δ(CDCl_3_) = 7.26 ppm) for ^1^H or the resonance signal (δ(CDCl_3_) = 77.16 ppm) for ^13^C. Coupling constants (*J*) are reported in Hertz (Hz), and the multiplicity is abbreviated as s (singlet), d (doublet), t (triplet), m (multiplet), br s (broad singlet), dd (doublet of doublets), ddd (doublet of doublet of doublets), ddt (doublet of doublet of triplets), dtd (doublet of triplet of doublets), and dddd (doublet of doublet of doublet of doublets). Signal assignment was performed with additional information of DEPT135, (^1^H, ^1^H)-COSY, (^1^H, ^13^C)-HSQC, and (^1^H, ^13^C)-HMBC. Atom numbers do not refer to the IUPAC nomenclature. High-resolution mass spectrometry (HRMS) was performed with a Bruker Daltonics micrOTOF and micrOTOF-Q III (electrospray ionization, ESI) instrument (Appendix A).

(a)tert-Butyl ((S)-6-azido-1-(((2S,3R)-1,3-dihydroxy-(E)-octadec-4-ene-2-yl)amino)-1-oxohexane-2-yl)carbamate 1

Under a dry N_2_ atmosphere, 6-azido-*N*-boc-*L*-norleucin (48.9 mg, 180 µmol, 1.3 eq.) was dissolved in dry dimethylformamide (DMF) (3 mL) and cooled to 0 °C. *N*,*N*-Diisopropylethylamine (69.8 µL, 401 µmol, 3.0 eq.) and (1-[Bis(dimethylamino)methylene]-1*H*-1,2,3-triazolo [4,5-b]pyridinium 3-oxide hexafluorophosphate (HATU) (64.5 mg, 170 µmol, 1.3 eq.) were added and the reaction stirred for 20 min. Then a solution of sphingosine (40.0 mg, 134 µmol, 1.0 eq.) in dry DMF (3 mL) was added. After 3 h, the reaction was neutralized with water (10 mL) and a sat. NH_4_Cl solution (20 mL) and the aqueous layer extracted with ethyl acetate (EtOAc) (3 × 25 mL). The combined organic layers were washed with brine (25 mL), dried over Na_2_SO_4_, and the solvent removed in vacuo. The crude product was purified by column chromatography (deactivated SiO_2_; CHCl_3_/MeOH, 90:1), which yielded a colorless solid (12.3 mg, 22.2 µmol, 45 %). R_f_ 0.13 (deactivated SiO_2_; CHCl_3_/MeOH, 80:1); ^1^H NMR (400 MHz, CDCl_3_): δ = 6.91 (d, 1H, *J* = 6.9 Hz, N*H*), 5.80 (dt, 1H, *J* = 7.0, 14.6 Hz, *H*-5), 5.52 (dd, 1H, *J* = 5.9, 15.5 Hz, *H*-4), 5.07 (d, 1H, *J* = 6.2 Hz, N*H*Boc), 4.36 (s, 1H, *H*-3), 4.03 (dd, 1H, *J* = 7.2, 13.5 Hz¸ *H*-2′), 3.97 (dd, 1H, *J* = 11.6, 3.1 Hz, *H*-1a), 3.83 (s, 1H, *H*-2), 3.73 (dd, 1H, *J* = 12.0, 2.1 Hz, *H*-1b), 3.29 (t, 2H, *J* = 6.7 Hz, H-6′), 2.05 (q, 2H, *J* = 7.1 Hz, *H*-6), 1.91–1.82 (m, 1H, *H*-3′a), 1.73–1.57 (m, 3H, *H*-3′b, *H*-5′), 1.53–1.44 (m, 2H, *H*-4′), 1.44 (s, 9H, *H*-3″), 1.38–1.33 (m, 2H, *H*-7), 1.31–1.25 (m, 20H, *H*-8–*H*-17), 0.88 (t, 3H, *J* = 6.8 Hz, *H*-18) ppm; ^13^C NMR (100 MHz, CDCl_3_): δ = 172.3 (*C*-1′), 156.2 (*C*-1″), 134.1 (*C*-5), 128.5 (*C*-4), 80.8 (*C*-2″), 74.2 (*C*-3), 62.0 (*C*-1), 55.3 (*C*-2′), 54.5 (*C*-2), 51.3 (*C*-6′), 32.5 (*C*-6), 32.1 (*C*-17), 31.8 (*C*-3′), 29.8–29.4 (8C, *C*-8–*C*-15), 29.3 (*C*-7), 28.6 (*C*-5′), 28.4 (3C, *C*-3″), 23.0 (*C*-4′), 22.8 (*C*-16), 14.3 (*C*-18) ppm; MS (ESI pos): *m*/*z* cal. for C_29_H_55_N_5_NaO_5_ [M + Na]^+^ 576.4095, found 576.4106, (|∆m/z| = 1.8 ppm) (Appendix A).

(b)(*S*)-2-Amino-6-azido-*N*-((2*S*,3*R*)-1,3-dihydroxy-(*E*)-octadec-4-ene-2-yl)hexanamide (α-NH_2_-ω-N_3_-C6-ceramide (AKS461))

**1** (30.0 mg, 54.2 µmol, 1.0 eq.) was dissolved in dry dichloromethane (DCM) (2 mL) and cooled to 0 °C. Trifluoroacetic acid (0.12 mL, 1.56 mmol, 29 eq.) was added, and the reaction was stirred for 2 h at this temperature. The reaction mixture was neutralized with water (5 mL) and 1 M NaOH (8 mL). The aqueous layer was extracted with EtOAc (5 × 15 mL). The combined organic layers were washed with brine (20 mL), dried over Na_2_SO_4_, and the solvent removed in vacuo. The crude product was purified by column chromatography (deactivated SiO_2_; DCM/MeOH, 20:1), which yielded a colourless solid (8.40 mg, 18.5 µmol, 34 %); R_f_: 0.08 (deactivated SiO_2_; DCM/MeOH, 20:1); ^1^H NMR (400 MHz, CDCl_3_): δ = 7.86 (d, 1H, *J* = 7.4 Hz, N*H*), 5.78 (dtd, 1H, *J* = 7.0, 14.9, 1.0 Hz, *H*- 5), 5.51 (ddt, 1H, *J* = 6.5, 15.4, 1.3 Hz, *H*-4), 4.30 (t, 1H, *J* = 4.9 Hz, *H*-3), 3.89 (dd, 1H, *J* = 11.9, 3.9 Hz, *H*-1a), 3.87–3.84 (m, 1H, *H*-2), 3.72 (dd, 1H, *J* = 10.9, 2.8 Hz, *H*-1b), 3.47 (dd, 1H, *J* = 5.2, 6.5 Hz, *H*-2′), 3.30 (t, 2H, *J* = 6.7 Hz, *H*-6′), 2.91 (s, 4H, N*H*_2_, 2 × O*H*), 2.05 (q, 2H, *J* = 7.0 Hz, *H*-6), 1.90–1.81 (m, 1H, *H*-3′a), 1.64–1.57 (m, 1H, *H*-3′b), 1.62 (q, 2H, *J* = 7.0 Hz, *H*-5′), 1.51–1.42 (m, 2H, *H*-4′), 1.38–1.33 (m, 2H, *H*-7), 1.31–1.25 (m, 20H, *H*-8-*H*-17), 0.87 (t, 3H, *J* = 6.9 Hz, *H*-18) ppm; ^13^C NMR (100 MHz, CDCl_3_): δ = 175.0 (*C*-1′), 134.5 (*C*-5), 128.7 (*C*-4), 74.1 (*C*-3), 62.4 (*C*-1), 55.1 (*C*-2), 55.1 (*C*-2′), 51.3 (*C*-6′), 34.4 (*C*-3′), 32.5 (*C*-6), 32.1–22.8 (10C, *C*-8–*C*-17), 29.3 (*C*-7), 28.8 (*C*-5′), 23.0 (*C*-4′), 14.3 (*C*-18) ppm; MS (ESI pos): *m*/*z* cal. for C_24_H_47_N_5_NaO_3_ [M + Na]^+^ 476.3571, found 476.3570, (|∆*m*/z| = 0.2 ppm) (Appendix A).

### 3.2. Viral Infection and RNA Quantification

The virus isolate has been described before [7,35]. The cells were seeded in 48 well plates (Vero, 15,000/well; Huh-7, 30,000/well; Calu-3, 100,000/well). Then the cells were incubated with AKS461 and, after approximately 15 min, infected with SARS-CoV-2 (MOI = 1). The medium was exchanged to remove inactive viruses, which influence genome copy determination. All infection experiments were performed in triplicate assays and repeated at least twice in independent experiments. After 72 h, 200 µL of the medium was collected, and viral genomes were purified with the High Pure Viral Nucleic Acid kit (Roche, Mannheim, Germany). SARS-CoV-2 RNA genomes were quantified with the dual-target SARS-CoV-2 RdRP RTqPCR assay kit, containing universal SARS-CoV-2 primers and with viral RNA multiplex master kit (Roche, Mannheim, Germany) with the LightCycler 480 II (Roche, Mannheim, Germany). The provided standard was used for genome copy-number quantification using the LightCycler 480 II Software Version 1.5 (Roche, Mannheim, Germany). The PCR reactions were performed in duplicates.

### 3.3. Cytotoxicity and Cellular Proliferation Assays

Huh-7 and Vero cells: The proliferation of cells was determined by direct automatic cell counting. Cells were seeded on optical plates (Vero, 3500/well; Huh-7, 5000/well) (CellCarier-96, PerkinElmer, Waltham, MA, USA) and counted before the experiments. The AKS461 was added at decreasing concentrations from 60 µM, and the cells were incubated for 72 h, like the infection time. The cell numbers per well were determined using the PerkinElmer Ensight reader. The numbers were compared to those of the solvent controls. All experiments were performed in six independent assays in parallel, and the standard deviation was calculated.

Calu-3 cells: Cytotoxicity of the AKS461 in Calu-3 cells was determined by MMT (3-[4,5-dimethylthiazol-2-yl]-2,5-diphenyl-tetrazolium bromide) assays (Promega, Mannheim, Germany). Calu-3 cells (20,000/well) were seeded in 96-well plates. The next day, AKS461 was added, and the cells were incubated for 72 h. Then, 10 µL of the MTT substrate was added, and the absorbance was measured after 1 h of incubation at 37 °C.

### 3.4. FISH Labelling

FISH of SARS-CoV-2 positive RNA-strand was performed as described [29]. Briefly, 93 primary oligos were ordered (MERCK KGaA, Steinheim, Germany) and mixed (oligonucleotide sequences see Appendix A). The mixture was diluted 5 times in TRIS-EDTA pH 8 (Sigma-Aldrich: 93283, St. Louis, MI, USA). For the labeling, a secondary dual-Cy5-conjugated imager-oligo was ordered at a concentration of 100 µM (MERCK KGaA) with sequence (5″ to 3″): [Cy5]AATGCATGTCGACGAGGTCCGAGTGTAA[Cy5]. Just before labeling of SARS-CoV-2-infected cells, the primary oligo solution was hybridized with the secondary imager-oligo solution in a thermocycler using the following composition and heating sequence: 2 µL primary oligo solution (20 µM) + 0.5 µL imager-oligo (100 µM) + 1 µL NEB3.1 buffer + 6.5 µL water (incubation at 85 °C for 3 min, at 65 °C for 3 min and finely at 25 °C for 5 min). Formaldehyde-fixed and 70% ethanol permeabilized cells were washed twice with 2× SSC buffer (Sigma-Aldrich: S6639), followed by two washing steps with 2× SSC supplemented with 10% formamide. The hybridized oligos were diluted in 500 µL hybridization buffer: 100 mg/mL dextran sulfate (Sigma-Aldrich: D8906) + 10% formamide (Sigma-Aldrich: F9037) in 2× SSC buffer. Cells were labeled overnight at 37 °C in a humidified and dark chamber. Cells were washed with 2× SSC + 10% formamide, PBS, and the nuclei were stained with 1 µg/mL Hoechst34580 (Sigma-Aldrich: 63493) for 15 min.

### 3.5. FISH Microscopy and Quantification

To ensure stable pH values, an imaging buffer (2× SSC, 50 mM Tris⋅HCl pH 8, 10% (*wt*/*vol*) glucose, 2 mM Trolox (Sigma-Aldrich), 0.5 mg/mL glucose oxidase (Sigma-Aldrich), 40 μg/mL catalase (Sigma-Aldrich)) was used for a maximum of 2 h after the addition of enzymes. LatticeSIM z-stacks were acquired with Elyra7 using a C-apochromat 63×/1.2 water immersion objective and 642 nm diode laser (150 mW, 5% laser power, 50 ms exposure time, 15 phases, 196 nm slicing, z-range ~10 µm) for excitation of Cy5 and Hoechst34580 was excited by the 405 nm diode laser. Detection and quantification of FISH clusters were performed on maximum-intensity projections with an ImageJ macro [29].

The lowest automatic threshold value used for the analysis of SARS-CoV-2-infected cells was also applied to non-infected control cells labeled by the FISH probes. Using this threshold (1100, 65,535), no single particle was detected in the maximum-intensity projections of control cells. Analysis of normal distribution, statistical significance, and graphical illustration of parameters particle area, circularity, and particle density was performed with OriginPro 2018b.

## Figures and Tables

**Figure 1 ijms-24-07281-f001:**
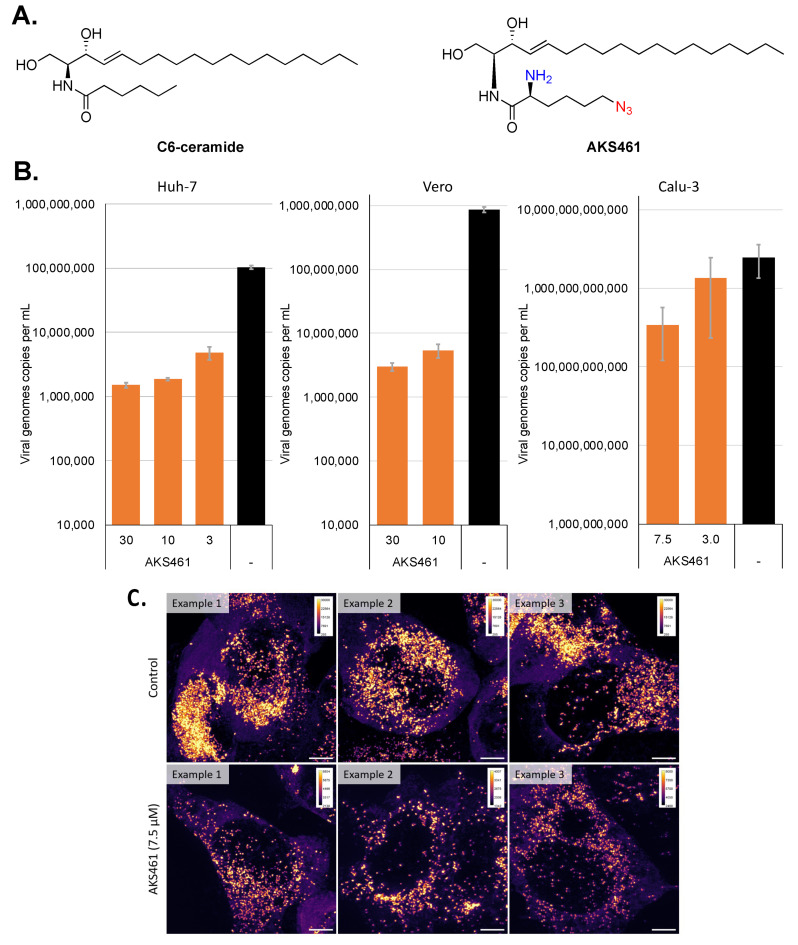
AKS461 inhibits SARS-CoV-2 replication. (**A**) Chemical structures of C6- and AKS461 ceramides. (**B**) Huh-7, Vero, and Calu-3 cells were incubated with AKS461 and infected with SARS-CoV-2 (MOI = 1). All infections were performed in triplicates. Viral loads were determined by RT-qPCR. (**C**) Intracellular SARS-CoV-2-RNA-FISH (three examples). Super-resolution SIM images of SARS-CoV-2 infected Vero cells labeled with CoronaFISH-Cy5.The CoronaFISH-Cy5 signal is reduced after treatment with 7.5 µM AKS461. Scale bar 5 µm.

**Figure 2 ijms-24-07281-f002:**
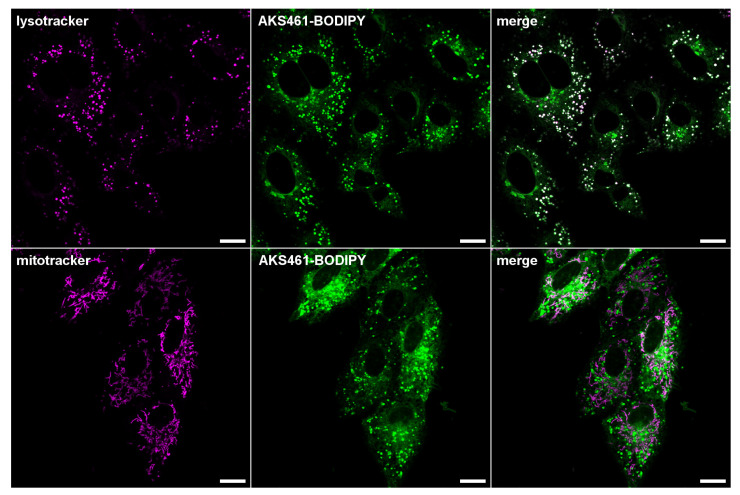
AKS461 is localized in the lysosome. Dual color live-cell visualization of AKS461 (**middle panel**) and lysosomes (**upper left panel**) or mitochondria (**lower left panel**). Vero cells were labeled with 50 nM LysoTracker Red DND-99 (**upper row magenta**) or 100 nM MitoTracker Deep Red (**lower row magenta**), and AKS461 clicked to DBCO-BODIPY (**green, middle panel**), respectively. Colocalization analysis using JACoP [34] showed a strong correlation between the signal of LysoTracker Red DND-99 and AKS461-BODIPY (Pearson coefficient 0.70 ± 0.02) and only a moderate correlation between MitoTracker Deep Red and AKS461-BODIPY (Pearson coefficient 0.47 ± 0.00). Scale bar 15 µm.

## Data Availability

Not applicable.

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
