# Peer review of "Azido-Ceramides, a Tool to Analyse SARS-CoV-2 Replication and Inhibition—SARS-CoV-2 Is Inhibited by Ceramides"

_ijms, 2023, doi:10.3390/ijms24087281_

Round 1

Reviewer 1 Report

It is undoubtedly an interesting communication that can be accepted under the following observations that the authors must comply with.

1.- Unless the editors approve it. Proponents must limit the number of corresponding authors and first authors and complement the affiliations according to the journal's instructions.

2.- Correct slight typing errors [e.g. SARS-CoV-2 2; (line 24)].

3.- Optional. In the Results and discussion section. The paragraph on lines 66-74) should be included and summarized in the Materials and Methods section.

4.- The authors must indicate the choice and variation of the use of concentration and incubation times in the various experiments (eg lines 82-83, 104,202).

5.- In figure 1B. You must specify in which volume the viral copy number is.

6.- In figure 1C. In the case of  signal intensity, could you indicate its intensity with graphs? Indicate in the figure caption that the representation of three experiments is treated at the same treatment concentration since it is not clear what each quadrant represents.  What does each quadrant represent? Specify in the figure itself

7.- Figure 2. What does each quadrant represent? Specify in the figure itself.

8.- Specify in the text the number of the corresponding figure in the section of supplementary materials (Supplementary materials) (lines 151, 227). Regarding this, in the text they only indicate “Supplementary materials” once and once “Supplementary information)” but in the corresponding section there are 4 figures. Make the pertinent corrections to the relationship between the mention of the figures in the text and in the supplementary information.

9.- They must indicate the multiplicity of infection used (MOI)  in the project.

10.- Indicate the choice of Cytotoxicity and cellular proliferation assays in the different cell lines.

11.- Supplementary information: line 227…. There is no additional information

Author Response

1.- Unless the editors approve it. Proponents must limit the number of corresponding authors and first authors and complement the affiliations according to the journal's instructions.

Our manuscript has been the joint work of three labs. That has been the reason for having three corresponding authors and 4 first authors. However, Markus Sauer decided to step down as the corresponding author. We would appreciate it if we could stay with this modified author list as an exception.

2.- Correct slight typing errors [e.g. SARS-CoV-2 2; (line 24)].

We corrected these errors.

3.- Optional. In the Results and discussion section. The paragraph on lines 66-74) should be included and summarized in the Materials and Methods section.

 We modified the manuscript accordingly.

4.- The authors must indicate the choice and variation of the use of concentration and incubation times in the various experiments (eg lines 82-83, 104,202).

We added this information.

5.- In figure 1B. You must specify in which volume the viral copy number is.

We added this information.

 6.- In figure 1C. In the case of  signal intensity, could you indicate its intensity with graphs? Indicate in the figure caption that the representation of three experiments is treated at the same treatment concentration since it is not clear what each quadrant represents.  What does each quadrant represent? Specify in the figure itself

 We added this information.

7.- Figure 2. What does each quadrant represent? Specify in the figure itself.

 We added this information.

8.- Specify in the text the number of the corresponding figure in the section of supplementary materials (Supplementary materials) (lines 151, 227). Regarding this, in the text they only indicate “Supplementary materials” once and once “Supplementary information)” but in the corresponding section there are 4 figures. Make the pertinent corrections to the relationship between the mention of the figures in the text and in the supplementary information.

 We modified the manuscript and added missing information to the supplementary information.

9.- They must indicate the multiplicity of infection used (MOI)  in the project.

We added this information. Lines 90, 126, 250

10.- Indicate the choice of Cytotoxicity and cellular proliferation assays in the different cell lines.

We added this information. Lines 262 & 270

11.- Supplementary information: line 227…. There is no additional information

 We modified the manuscript and added missing information to the supplementary information.

Reviewer 2 Report

The authors in the Communication “Azido-ceramides, a tool to analyse SARS-CoV-2 replication and  inhibition – SARS-CoV-2 is inhibited by Ceramides” presente a new derivative of C6-ceramide modified to can used in microscopy studies and confirm the hypothesis that the ceramides can to inhibit SARS-CoV-2 replication.  The article is interesting and well presented, the experiments are well designed and clearly presented. I recommend the pubblication of the article on International Journal of molecular science after major revisions.

-         - Vero cells were treated with 7.5 µM AKS461, infected with SARS-CoV-2, and viral RNA visualized using super-resolution structured illumination microscopy (SIM) The results indicate that AKS461 has a similar impact on viral replication as C6-ceramide. Do you tryed to visualized the cells over time (time 72 > hours) and to use an increase concentration of AKS461?

-         - The authors perform the experiments in a time scale of 72 hours! Do you have informations on serum stability of AKS461?

-         - Line 66-67 the authors state that “the purification of AKS461 is challenging”. What are the limitations encountered and what were been the improvements achieved? 

-          - In the Figure 1B add the unit of measure at the AKS461 reported on the abscissa axis.

Author Response

Vero cells were treated with 7.5 µM AKS461, infected with SARS-CoV-2, and viral RNA visualized using super-resolution structured illumination microscopy (SIM) The results indicate that AKS461 has a similar impact on viral replication as C6-ceramide. Do you tryed to visualized the cells over time (time 72 > hours) and to use an increase concentration of AKS461?

We did not perform a time-course experiment. However, we performed refeeding experiments with C6 Cermides, which did not result in significant changes in the inhibition of SARS-CoV-2

- The authors perform the experiments in a time scale of 72 hours! Do you have informations on serum stability of AKS461?

Interesting question, but we have no information on that.

-         - Line 66-67 the authors state that “the purification of AKS461 is challenging”. What are the limitations encountered and what were been the improvements achieved? 

The main problem has been the toxicity of the sphingosine educt, which required a better purification scheme. We added this information to the manuscript.  

-          - In the Figure 1B add the unit of measure at the AKS461 reported on the abscissa axis.

We added this information